# Molecular elucidating of an unusual growth mechanism for polycyclic aromatic hydrocarbons in confined space

Nan Wang [1,2,5], Yuchun Zhi[1,5], Yingxu Wei[1✉], Wenna Zhang[1], Zhiqiang Liu [3], Jindou Huang[1], Tantan Sun[1,2], Shutao Xu [1], Shanfan Lin[1,2], Yanli He[1], Anmin Zheng [3] & Zhongmin Liu [1,4✉]

Extension and clustering of polycyclic aromatic hydrocarbons (PAHs) are key mechanistic steps for coking and deactivation in catalysis reactions. However, no unambiguous mechanistic picture exists on molecule-resolved PAHs speciation and evolution, due to the immense experimental challenges in deciphering the complex PAHs structures. Herein, we report an effective strategy through integrating a high resolution MALDI FT-ICR mass spectrometry with isotope labeling technique. With this strategy, a complete route for aromatic hydrocarbon evolution is unveiled for SAPO-34-catalyzed, industrially relevant methanol-to-olefins (MTO) as a model reaction. Notable is the elucidation of an unusual, previously unrecognized mechanistic step: cage-passing growth forming cross-linked multi-core PAHs with graphene-like structure. This mechanistic concept proves general on other cage-based molecule sieves. This preliminary work would provide a versatile means to decipher the key mechanistic step of molecular mass growth for PAHs involved in catalysis and combustion chemistry.

---

[1] National Engineering Laboratory for Methanol to Olefins, Dalian National Laboratory for Clean Energy, iChEM (Collaborative Innovation Center of Chemistry for Energy Materials), Dalian Institute of Chemical Physics, Chinese Academy of Sciences, Dalian 116023, PR China. [2] University of Chinese Academy of Sciences, 100049 Beijing, PR China. [3] National Center for Magnetic Resonance in Wuhan, State Key Laboratory of Magnetic Resonance and Atomic and Molecular Physics, Key Laboratory of Magnetic Resonance in Biological Systems, Wuhan Institute of Physics and Mathematics, Innovation Academy for Precision Measurement Science and Technology, Chinese Academy of Sciences, Wuhan 430071, PR China. [4] State Key Laboratory of Catalysis, Dalian Institute of Chemical Physics, Chinese Academy of Sciences, Dalian 116023, PR China. [5] These authors contributed equally: Nan Wang, Yuchun Zhi. ✉email: weiyx@dicp.ac.cn; liuzm@dicp.ac.cn

Polycyclic aromatic hydrocarbons (PAHs) are ubiquitous, ranging from combustion chemistry to catalysis chemistry. The chemistry of PAHs growth remains a long-pursued yet largely unresolved scientific puzzle, such as in the fields of incomplete combustion of fuels as well as hydrocarbon and biomass hydrolysis (wherein human body- and environment-detrimental soot are formed)[1–3]. The PAHs growth in zeolite-catalyzed chemical reactions would be analogous to soot formation, as both processes refer to the extension and clustering of preliminarily formed aromatic molecules. But, the PAHs growth in catalysis process may display distinct features in that nanosized pores of zeolites would regulate the dimensions and structural motifs of PAHs.

The formation of PAHs is inevitable in catalysis processes, especially in industrially important petrochemical processes (catalytic cracking, isomerization, transalkylation of aromatics, etc.) and coal-based chemical processes (such as methanol-to-olefins (MTO) and syngas conversion). The deposition of these extended aromatic hydrocarbons detrimentally induce catalyst coking and deactivation, necessitating regeneration operations, and posing challenges to practical process design. To alleviate or even prevent catalyst deactivation, unambiguously unraveling of the structural motifs of PAHs, at a molecular scale, is necessary. Despite enormous endeavors, a full-molecular description of the precise chemical structures of these PAHs is far from being achieved. Yet, if a full-spectrum deciphering of such PAHs is realized, it would become feasible to trace and uncover the molecular route of PAHs formation and evolution, and thus to obtain unprecedentedly detailed structural information of coke deactivating the zeolite. This would, in turn, help in improving industrial processes. Hence, this work aims to definitely fingerprint the molecular identity of those complex PAHs.

Tremendous efforts have been made to identify and locate PAHs in catalytic reactions. The well-established Guisnet's method[4], combining inorganic acid dissolution, solvent extraction, and gas chromatograph-mass spectrometry (GC-MS) analysis, can identify and quantify a large proportion of the retained organics. Nevertheless, this method is limited to the analysis of molecules with maximum molar weight below $300 \, g \, mol^{-1}$, due to the decreased solubility with increasing size of PAHs[5]. Various sophisticated spectroscopic techniques such as infrared spectroscopy[6], UV-Raman spectroscopy[7–9], UV-Vis spectroscopy[10], and $^{13}C$ nuclear magnetic resonance (NMR) spectroscopy[11–13], with the merits of being nondestructive and operable in situ or operando modes, have also been extensively used. These spectroscopic techniques are capable of differentiating the insoluble, highly condensed PAHs from the soluble, light constituents, yet, they fail in determining the molecular structure of the insoluble fraction of coke[14]. Although the multitechnique approaches have been further advanced[10], it is still rather difficult to establish the detailed chemical composition of PAHs.

Of note is the pioneering work contributed by Weckhuysen et al. that provides critical insights into coking and deactivation. With the help of two advanced microscopic techniques: confocal fluorescence microscopy (CFM)[10,15] and atom probe tomography (APT)[16–18], they visualized the spatial distribution of coke deposits at sub-μm or sub-nm scale in a single catalyst particle, and identified the affinity between the acid sites (enriching on the near-surface region of catalysts) and the coke clusters (located in the same region)[16,17]. These microscopic techniques can unprecedentedly provide effective information on the spatial distribution of coke molecules. They are, however, still unable to provide molecular fingerprints of those clustered PAHs.

Fortunately, sophisticated mass spectrometry (MS) techniques, such as tandem MS[1], HPLC/UV/MS[19], and synchrotron vacuum ultraviolet photoionization MS[20], have been employed for identifying the molecular structure of PAHs precursor to form soot or carbonaceous deposits during hydrocarbon combustion and pyrolysis processes. However, given the diverse isomers of each specified mass, accurate molecular structures are rather difficult to be fingerprinted, and their structural confirmation normally necessitates reference standards or pre-determined/published UV spectra[19,21]. Noticeably, among these MS techniques, matrix-assisted laser desorption/ionization Fourier-transform ion cyclotron resonance mass spectrometry (MALDI FT-ICR MS) (that was traditionally applied for deciphering the structure of biological molecules) has emerged as a powerful tool for coke structural identification in zeolite-catalyzed reactions with the advantage of soft ionization thus minimizing the fragmentation[22]. Only few studies are available to date that used MALDI FT-ICR MS to identify the nature of coke species over HBEA[23], HMOR[24], and HZSM-5[17], but this tool alone was unable to address the complexity of isomer-specific measurements.

Herein, we present an effective strategy, by coupling the MALDI FT-ICR MS with isotope labeling (Fig. 1), to probe the molecular-scale structural fingerprints of larger PAHs molecules, overcoming the intrinsic structure-discriminating limit confronted by the conventional spectroscopic and microscopic techniques. Taking industrially important SAPO-34-catalyzed MTO reaction as a model reaction, we first rule out the possibility of appreciable coke situated on the external surface of SAPO-34, and then unveil how these bulky PAHs molecules are accommodated in small cavities. By identifying the structural fingerprints of larger PAHs molecules, we propose a PAHs cage-crossing growth mechanism which is later proved general on other cage-structured molecular sieves. This work advances the long-pursued structural identification of larger PAHs molecules. The complementary analysis approach allows us to assemble a full picture of coke analysis in industrially relevant methanol conversion reaction (Fig. 1). The newly developed strategy is expected to be a versatile approach for unraveling PAHs growth mechanism in other catalytic reactions and other fields.

## Results

**Choosing industrially important MTO as a model reaction.** The molecular-level mechanism of PAHs growth and the concomitant coking and deactivation remarkably depend on the zeolite framework[25] and the operating conditions such as reaction temperatures, pressures, and nature of the reactants[26,27]. In this context, some conclusions reached under certain reaction conditions might not be applicable to the cases under other conditions. For instance, the dominating deactivating species for SAPO-34-catalyzed MTO reaction evolve from adamantanes at around 270–300 °C[28,29], to methylated naphthalene at around 300–450 °C, and then to more condensed PAHs when temperature is above 450 °C[30]. The molecular route of PAHs at high temperature (usually above 450 °C) is highly industrially relevant, under which MTO process suffers from quick catalyst deactivation. The quick deactivation is related to the topological structure of SAPO-34 with a large cage ($10.9 \, Å \times 6.7 \, Å$) and small pore opening ($3.8 \, Å \times 3.8 \, Å$) imprisoning aromatics and even branched paraffins as deactivating species. The topological architecture of zeolite or molecular sieve imparts unique confined space to the guest molecules such as PAHs, so that the speciation and developing of such "coke" molecules would be regularly dictated in a controllable manner in confined space. In this regard, the MTO reaction taking place on cage-structured SAPO-34 can be an ideal probe to trace the mechanistic routes of PAHs.

The accumulation of "coke" (including PAHs) with reaction progress can gradually enhance ethene selectivity on SAPO-34[31,32].

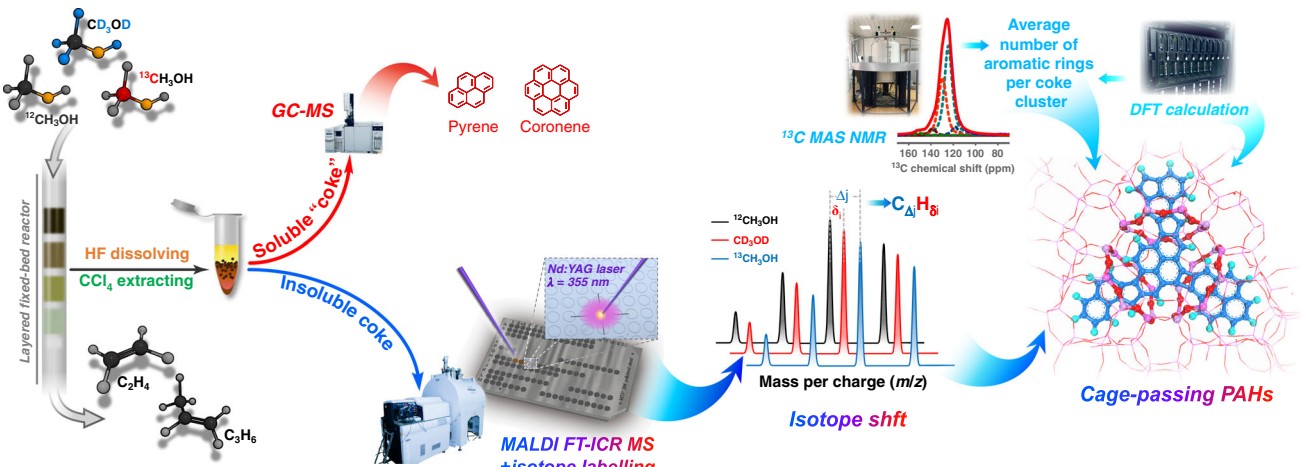

**Fig. 1 Schematic illustration of an integrated strategy for deciphering full-spectrum PAHs.** The proposed analytic method for heavier PAHs by combining the advanced MALDI FT-ICR MS with [13]C magic-angle spinning nuclear magnetic resonance (MAS NMR) spectroscopy and density functional theory (DFT) calculations. This method, together with the improved Guisnet's method (see "Methods"), allows for a full-spectrum analysis of coking species ranging from soluble "coke" to insoluble heavier PAHs. For the cage-structured zeolite- or molecular sieve-catalyzed reactions, certain amounts of branched hydrocarbons and active aromatic species can also be retained inside the cages of catalyst after reaction. So, in order to avoid misleading, we herein term the soluble "coke" being the organics dissolved in $CCl_4$ solvent and detectable by GC-MS, and the residual heavier counterpart being insoluble coke.

Inspired from this counter-intuitive but suggestive phenomenon, pre-coking (pre-filling SAPO-34 cages with certain PAHs before reaction)[32] and partial regeneration (partially burning-off the carbonaceous deposits) techniques have been developing among the MTO industry to modulate ethene selectivity. In this context, the chemical nature and molecular pathway of PAHs growth involved in these three critical processes (catalyst deactivation, pre-coking, and partial regeneration) is not only of academic interests, but also of a great industrial significance. Although the nature of lighter PAHs (soluble "coke") has been well understood, the definite chemical nature of heavier PAHs (insoluble coke) and their evolution route from the soluble "coke" are still unclear. With all these points in mind, we herein chose MTO reaction as a prototypical case, aiming to identify the structural fingerprints of the extended, heavier PAHs and then to elucidate their molecular route.

Noteworthily, spatiotemporal inhomogeneity of coking (i.e., inhomogeneous axial deactivation along the catalyst bed)[33,34] augments the uncertainty of sampling for coke analysis, thus hampering the accurate extraction of structural information from PAHs species. To reduce the effect of inhomogeneous axial deactivation within control, when performing MTO reactions we separated the catalyst bed equally into four layers using quartz wool (Supplementary Fig. 1) as described in "Methods".

**Catalyst analysis and general aspects of MTO reaction.** The layer-by-layer coking phenomenon appeared clearly for MTO reaction over SAPO-34 (see physicochemical properties in Supplementary Fig. 2 and Supplementary Tables 1 and 2) at 475 °C and weight hourly space velocity (WHSV) of $4\,h^{-1}$ (Supplementary Fig. 1). With reaction proceeding, the reaction zone gradually migrated downward. It should be noted that in most of the following tests only spent catalysts from the top layer were analyzed and discussed, in view of secondary reactions of the product olefins prevailing within the bottom layer of the catalyst bed, especially at the early stage of this reaction[35]. The catalytic performance of methanol conversion is presented in Fig. 2b. Typically, ethene and propene dominated the effluent products. $C_4^+$ products gradually declined in parallel with the increase of

methane, being typical after around 55 min of reaction (that is, after pseudo steady-state stage). Methanol started to appear in the effluent after 55 min of reaction, implying that the fraction of accessible SAPO-34 pores dropped below a critical threshold, unable to sustain full methanol conversion[36]. During the pseudo steady-state stage, both the specific microporous surface area and pore volume were significantly reduced by ~80% (Supplementary Table 1) associated with the almost linear increase of the deposited "coke" (Fig. 2c), signifying the fast "coke" filling inside the SAPO-34 pores. After that, the decrease in microporous surface area and pore volume turned slow, in line with the slow increase in "coke" buildup.

The carbonaceous deposits were liberated by the modified hydrofluoric acid (HF) dissolution-$CCl_4$ extraction method (see "Methods"). This method previously proved not to cause any significant damage to the carbonaceous deposits[4]. GC-MS analysis of the extracted soluble "coke" from the top-layer catalysts shows that the retained aromatic species (from polymethylbenzenes to pyrene) continued to increase in abundance with time on stream (TOS), further shifting toward heavier constituents (Fig. 2d). The proportion of soluble "coke" decreased monotonously with TOS, while the insoluble coke increased for the top-layer catalysts. This also holds true for the bottom-layer and four-layers mixed catalysts, except for the initial stage (after 5 min of reaction) (Supplementary Fig. 3). These observations imply that insoluble coke stems from the extension and clustering of soluble "coke" (Fig. 2a). The prevalence of insoluble coke suggests its pivotal role in the catalytic process, which is, however, previously under-explored. Thus, to obtain a full picture of the deactivation mechanism, special attention should be paid to the molecular structure and evolution of the insoluble coke. Before probing the nature of such heavier PAHs, we first discern the spatial location of these "coke" moieties.

**Ruling out the possibility of appreciable on-surface coking.** Ångstrom-sized *cha* cages of SAPO-34 sterically restrict the extent of growth (or the dimension) of the occluded aromatic molecules. We first estimated by density functional theory (DFT) calculations the maximum size of aromatic molecules that can be

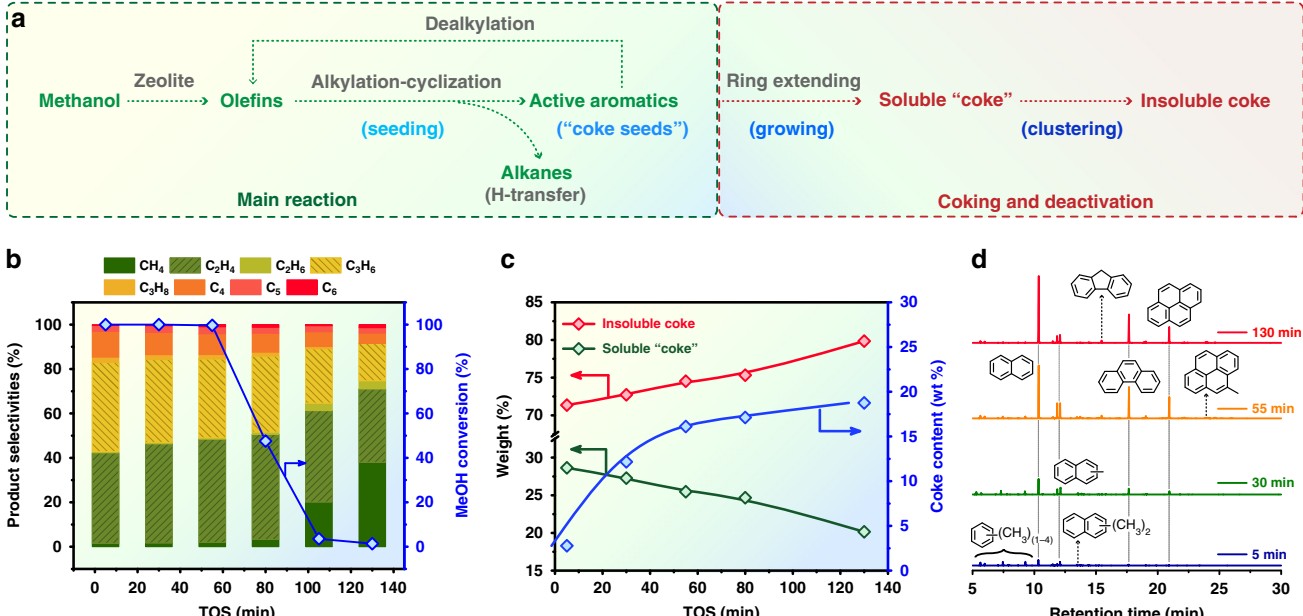

**Fig. 2 Catalytic performance of methanol conversion and preliminary "coke" analysis. a** The whole evolving route of hydrocarbon pool species during the MTO process under industrially relevant high temperature conditions. The coking and deactivation go through three stages: seeding (forming methylbenzene species as "coke seeds"), growing (extending aromatic rings to PAHs), and clustering (cross-linking of PAHs as discussed later). **b** MTO reaction performance over SAPO-34 at 475 °C and WHSV of 4 h$^{-1}$. **c** Total coke content determined by thermogravimetric analysis (TGA) with time on stream (TOS) and the weight percentage of soluble "coke" and insoluble coke by analyzing the samples in the up layer of catalyst bed. The amount of soluble "coke" was determined by summation of the aromatic extracts dissolved in CCl$_4$ with C$_2$Cl$_6$ as an internal standard for quantification. Difference between the total "coke" and soluble "coke" yields the insoluble coke amount. **d** Time-evolving chromatograms of CCl$_4$ soluble fractions of the retained organics (soluble "coke") extracted from the spent up-layer catalysts after HF digestion.

accommodated in one *cha* cage. The calculation results show that the condensed aromatic molecules larger than pyrene are energetically unstable and unlikely to be formed in *cha* cage, on grounds of the curved configuration and repulsive interactions with cage wall (evidenced by the positive adsorption energy above 110 kJ mol$^{-1}$) for the five- to six-ring analogs (Supplementary Fig. 4). This theoretical estimation agrees well with the experimental observation of pyrene and its derivatives being the maximum-sized aromatic hydrocarbons in the GC-MS analysis (Fig. 2d) and the previous reports[37,38]. The lattice expansion of ca. 1–3% in *c*-axis direction of SAPO-34 crystallite during the MTO reaction at 400–500 °C (due to the filling of active intermediates and PAHs) was revealed by in situ X-ray diffraction (XRD) [39,40]. Although the deposited coke slightly alters the unit cell parameters of SAPO-34 crystallite, we surmise such marginal and insignificant variations would not have appreciable effect on the accommodated aromatic molecules.

The theoretical and experimental demonstrations of pyrene being the maximum-sized PAH occluded in a separate *cha* cage lead us to speculate on whether the substantial insoluble coke is located on the external surface of SAPO-34. Herein, we can exclude any catalytically consequential extents of external surface coking based on the following chain of arguments: (1) Less C$_6^+$ hydrocarbons and no detectable aromatics were present in the effluent products (Fig. 2b), suggesting rather weak or negligible acidic sites on the external surface (non-selective reactions would occur otherwise)[41]; (2) The rather low outer surface area (below 10 m$^2$ g$^{-1}$ accounting for <2% of the total surface area) associated with the low mesopore volume of the applied SAPO-34 catalyst (Supplementary Table 1) makes it unlikely that the on-surface coking has occurred to a noticeable extent; (3) The almost linear correlation between the reduced micropore volume and the increasing "coke" content (Supplementary Fig. 5) indicates the

selective filling of "coke" inside the micropore pores, further negating the possibility of considerable on-surface coking; (4) More solid direct evidence for the rather less extent of on-surface coking was delivered by the advanced MALDI FT-ICR MS. The deactivated SAPO-34 samples (without dissolving the framework by HF) were directly tested by MALDI FT-ICR MS to probe the possibility of carbonaceous residues deposited on the external surface. The similar MS method was applied on HZSM-5[17] and HBEA[23] and the results showed massive coke molecules detected on the external surface. But, the same is not true for SAPO-34, as the MS results demonstrated no remarkable MS peak signals of carbonaceous deposits occurring on the external surface of SAPO–34 with different reaction or deactivation extents (Supplementary Fig. 6). It should be emphasized that the above arguments do not mean to totally exclude the occurrence of external-surface coke; on-surface coking may occur, but to a rather small extent. The dominance of in-pore coking for SAPO-34 contrasts the external coking for HZSM-5[10,17], largely due to the topological differences (cavity-structure vs. channel-structure).

Hitherto, few works detailed the structural identity and spatial location of insoluble coke on SAPO-34. Rostami et al.[37] and Konnov et al.[42]. speculated that the insoluble coke species was formed on the outer surface (inter-crystalline voids). However, this argument lacked convincing experimental support and was merely deduced from the observation that *cha* cages of spent catalysts hardly contain aromatic molecules larger than pyrene. Instead, Weckhuysen et al.[10] probed the coke precursor and deposits by in situ spectroscopy and visualized the clustered PAHs being preferentially formed on the outer-shell cages of SAPO-34, in support of our observations.

As analyzed above, the possibility of appreciable external surface coking has been eliminated. This conclusion allows us to infer that, with high confidence, the major fractions of extended

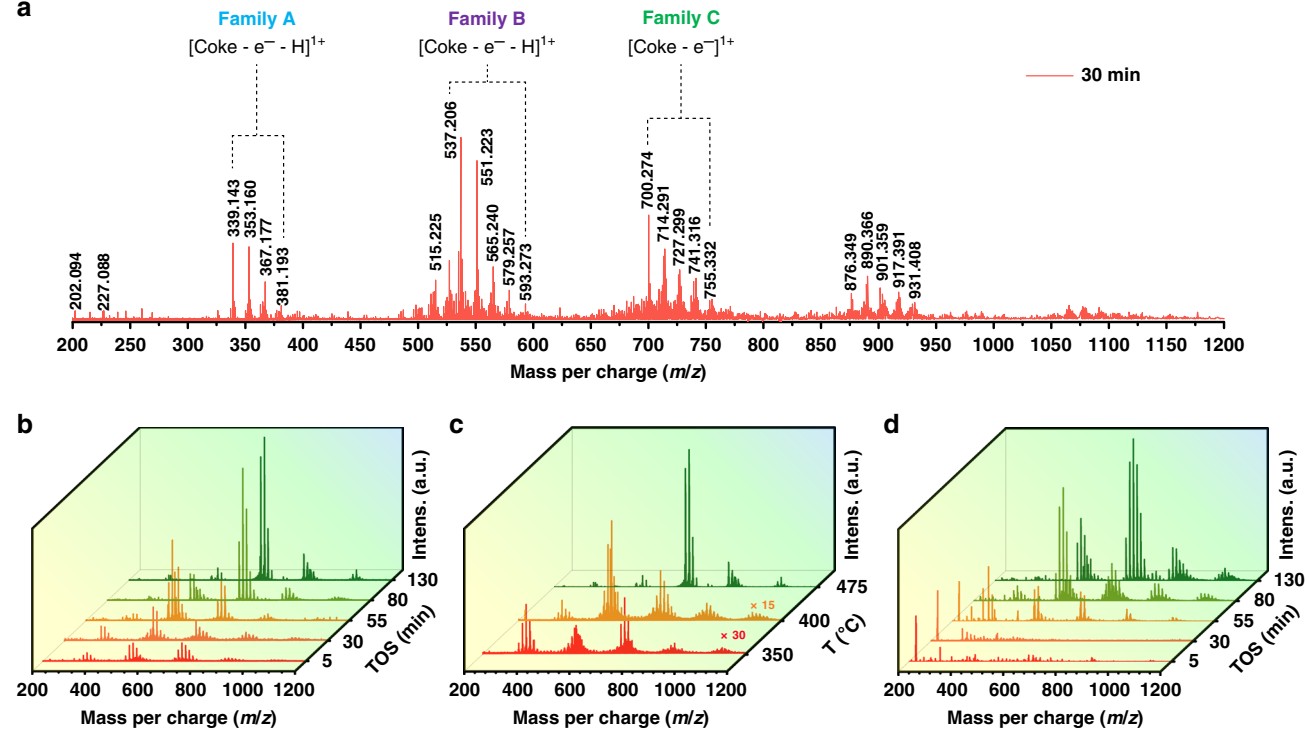

**Fig. 3 MALDI FT-ICR mass spectra of the liberated carbonaceous deposits.** The mass spectra of the liberated carbonaceous deposits obtained from the used up-layer SAPO-34 catalysts by HF dissolution-CCl₄ extraction after MTO reaction with WHSV of 4 h⁻¹ at 475 °C for 30 min (**a**), at 475 °C for different durations (5–130 min) (**b**), at different temperatures (350–475 °C) (**c**). **d** The mass spectra of the liberated carbonaceous deposits obtained from the used bottom-layer catalysts after MTO reaction at 475 °C for different durations (5–130 min).

PAHs should situate inside the cages of SAPO-34. This reasoning raises the question of how such bulky polyaromatic compounds can be suitably accommodated in nanosized *cha* cages. To answer this question, we next adopted MALDI FT-ICR MS to fingerprint these extended PAHs.

**Structural features of heavier PAHs**. Prior to carbonaceous deposit measurements, a series of control experiments were conducted, using dithranol matrix, model compound (9,10-di-(2-naphthyl) anthracene) and extracts from the spent SAPO-34 catalyst as standard and reference samples (Supplementary Fig. 7), to achieve the optimal MS parameters for our sample system.

Subsequently, the liberated carbonaceous deposits in extracts were analyzed by MALDI FT-ICR MS. The mass distribution of the detectable coke species ranged from 300 to 1200 Da (Fig. 3a–d), which succeeds the upper limit of mass given by traditional GC-MS method. As such, combining MALDI FT-ICR MS and GC-MS has the potential to offer a full-spectrum interpreting of "coke" structural information. An unusual and interesting observation for the SAPO-34-derived coke species was the mass profile being piecewise distributed in a discrete manner. The spectra were composed of several fragment ion groups in *m/z* ranges of 320–400, 500–600, 680–780 Da, etc. Each group features Gaussian-curved distributions of masses with CH₂ (14 Da) increment (substitution of H– by CH₃– group) and maxima at 339, 537, and 700 Da, etc. (the spectrum shown in Fig. 3a for illustrative purposes). Such grouped, disconnected mass spectral patterns again suggest these heavier insoluble aromatics are unlikely to be situated on the outer surface of SAPO-34, in that, if so, the *m/z* peaks should appear continuous, just like the cases for HZSM-5[17] and HBEA[23] wherein light "coke" species tend to

diffuse out and evolve into graphite-like coke on the outer surface.

Of particular note is the regular interval of around 164–214 Da between the neighboring fragment groups, which corresponds to the mass range of three- to four-ring aromatic species, e.g., fluorene (166), phenanthrene (178), and pyrene (202). The similar observations were also found under lower temperature conditions, for example, 350 and 400 °C, though with apparently weak spectral peaks indicating less insoluble coke formed (Fig. 3c). This was also the case for the bottom-layer catalysts (Fig. 3d), except for the spectra in the initial reaction stage (the first 30 min) where little insoluble coke was formed and no typical Gaussian-curved distributions of masses occurred, a result of the prevailing reaction of relatively less reactive olefins (relative to methanol)[43]. Such mass distribution feature (with three- to four-ring aromatic mass interval), associated with the theoretical estimation of pyrene being the maximum-sized molecule contained in one *cha* cage, permits us to hypothesize that the bulky PAHs (not likely to sit in one *cha* cage) are prone to occupy several proximal cages by traversing the neighboring eight-membered ring windows. This hypothesis is intuitively surprising but is plausible in view of the CHA structure with one cage being accessible to adjacent six ones. We can conceive that three- to four-ring aromatic hydrocarbons are firstly formed in the separate cages and behave as the primary coking units/clustering seeds, and then they are gradually cross-linked with the ones in the neighboring cages. This process enlarges the size of aromatic molecules through the successive addition of three- to four-ring aromatic molecules via cage-passing growth, well explaining the discrete, grouped feature of measured MALDI FT-ICR mass spectra.

**The structural identification of PAHs by a combined strategy**. To testify this hypothesis, we designed the following more

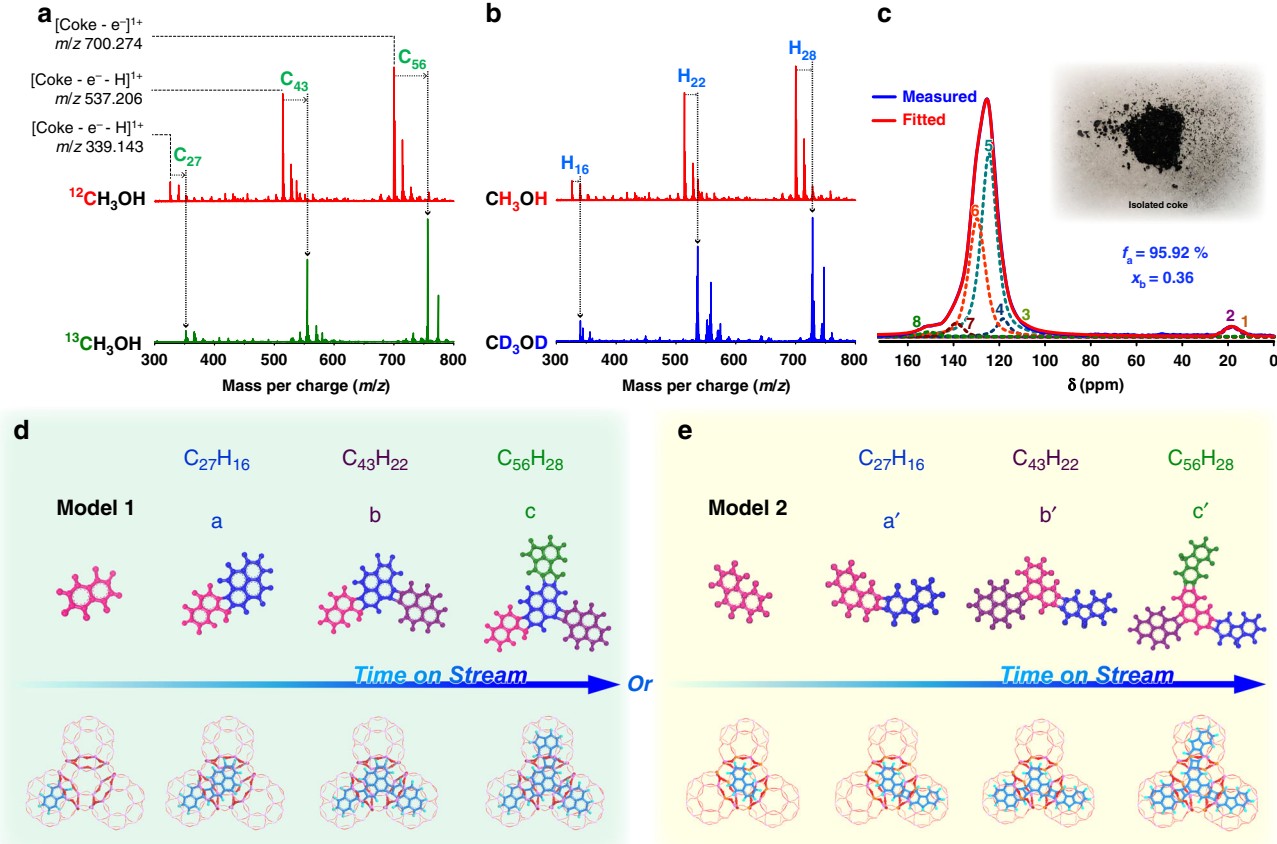

**Fig. 4 The structural identification of PAHs.** MALDI FT-ICR mass spectra of the extracts from the spent SAPO-34 after 130 min reaction at 475 °C for comparing the feeding of $^{13}$C-methanol (**a**) and D-methanol (**b**) with normal methanol. **c** $^{13}$C solid-state nuclear magnetic resonance (ssNMR) spectroscopy of the isolated insoluble coke. The spectrum was recorded at a spin rate of 110 kHz. The morphology of isolated coke was shown in Supplementary Fig. 8. **d, e** The identified conceivable molecular structures of PAHs with optimized configuration (Model 1 and Model 2) by periodic DFT calculations and their evolving routes. The configurational optimization was shown in Supplementary Fig. 9. Although with the precise $C_x$ (**a**) and $H_y$ (**b**) that give the precise chemical composition of each PAH ($C_xH_y$), the diversity of isomers would lead to multiple molecular structure. Fortunately, in view of the connectivity and dimensions of *cha* cages, some different isomers have been excluded. Here we stress that the illustrative examples of molecular structure are certainly not absolute, and they highlight the concept of cage-passing growth of PAHs.

convincing experiments by coupling MALDI FT-ICR MS with isotope labeling. Specifically, feeding $^{13}$C-methanol as reactant would shift the mass pattern to larger masses; the resulting mass increment for each maximum peak relative to that for $^{12}$C-methanol corresponds to the accurate carbon atom number ($C_x$) of the corresponding PAH. By the same token, when deuterated methanol (abbreviated as D-methanol) was fed as reference to normal methanol, the corresponding hydrogen atom number ($H_y$) can also be determined. The attained accurate chemical compositions ($C_xH_y$) can, however, be matched with multiple molecular structures in view of the diversity of isomers. Fortunately, the compositional differences between each PAH ($C_xH_y$) with the preceding one ($C_{x-m}H_{y-n}$) can explicitly deduce the chemical composition of each building unit ($C_mH_n$) (Supplementary Table 3), with which we can narrow down the isomers to rather limited ones.

Recall that the reaction durations and temperatures affect the mass pattern of condensed PAHs—quantitatively not qualitatively (Fig. 3a–d). Hence, herein we collected the extracts from the upper-layer samples after 130 min of reaction at 475 °C as a probe to establish the potential chemical structures of PAHs. $^{13}$C-methanol (Fig. 4a) or D-methanol (Fig. 4b) feeding incrementally shifted the mass patterns (relative to normal methanol feeding), with $C_x$ increments of 27, 43, and 56 and $H_y$ of 16, 22, and 28 for the respective prominent peaks. With this, the chemical

composition of these maximum mass peaks can be ascertained to be $C_{27}H_{16}$, $C_{43}H_{22}$, and $C_{56}H_{28}$. They are the representatives of those clustered-mass groups with the general formulas of $C_nH_{2n-40}$ ($27 \leq n \leq 31$), $C_nH_{2n-66}$ ($43 \leq n \leq 48$), and $C_nH_{2n-84}$ ($56 \leq n \leq 61$), respectively, indicating the different extents of $CH_2$ (14 Da) addition (Supplementary Table 3). Notably, the unsaturated number (UN) for these coke molecules increased by 20, 33, and 43 (Supplementary Table 3), manifesting coke growth by ring condensation, contrary to the step-by-step growth with successive UN occurring on medium- and large- pore zeolites, such as HZSM-5[17], HBEA[23], and mordenite[24]. In the context of *cha* cage structure and dimension as well as the above-inferred coke building units (three- to four-ring aromatics), we reason that $C_{27}H_{16}$ may display the possible molecular structure of **a** or **a′**, occupying two cages (Fig. 4d, e). The mass differences between $C_{27}H_{16}$ and $C_{43}H_{22}$ ascertain pyrene as the third added structural unit into **a** or **a′** (Supplementary Table 3), giving the molecular structure of **b** or **b′** which extends into three cages. By analogy, we can deduce the fourth added structure unit (three-ring aromatic) and the further extended molecules of **c** or **c′** that passes through four cages. We envision that more extended PAHs may exist, with the above-identified, repeated three- to four-ring aromatic structural units, given that these detected PAHs species were spectroscopically probed within the detection/ionization limit of our used MALDI FT-ICR MS. The identified molecular

structures of PAHs shown in Fig. 4d, e also manifest their molecular evolving route, representing a novel proposal of coke buildup mode via cage-passing ring condensation. By the aid of DFT-based configurational optimization, these biphenyl- and/or methylene-bridged multicore PAHs present single-layer graphene-like three-dimensional configurational structures (Fig. 4d, e). Noteworthily, these preliminary findings are meant to highlight the interesting cage-passing structural feature of multicore PAHs, albeit other conceivable molecular structure can also be expected.

$^{13}$C ssNMR was a powerful technique capable of identifying the fine carbon structure in PAHs, for example, crude oil components[11,12] and zeolite-templated carbon material[44]. Next, we use isolated $^{13}$C-labeled coke (which was obtained by suction filtration of the extracts by feeding 99.9% $^{13}$C-enriched methanol; see "Methods") for $^{13}$C ssNMR test to further testify the above-identified molecular structural features of insoluble coke. The same mass distributions (confirmed by MALDI FT-ICR MS; see Supplementary Fig. 10) of the extracts with the isolated coke make this $^{13}$C ssNMR analysis feasible and convincing. $^{13}$C ssNMR results are shown in Fig. 4c. The aliphatic carbon bands (0–90 ppm)[11] was rather weak, while aromatic carbon bands (90–160 ppm)[11] dominated the $^{13}$C NMR spectrum, suggesting the $sp^2$ hybridized nature of such graphene-like coke. With reference to the reported curve-fitting analysis[11,12], the NMR spectrum can be further fitted into eight individual peaks assigned to different types of aromatic and aliphatic species (Fig. 4c and Supplementary Fig. 11). The less aliphatic carbon being mainly comprised of aromatic methyl ($f_{al}^a$) is consistent with the low substitution degree ($\sigma$) of aromatic rings (defined as $\sigma = f_a^S/f_a$,

calculated to be 0.037. The high aromaticity ($f_a$) of up to 95.9% suggests the aromatic nature of the majority of carbon atoms. The molar percent of aromatic bridgehead carbon ($x_b$), an indicator for aromatic cluster size[45], was estimated to be 0.36, in between the value of phenanthrene (0.286) and pyrene (0.375). With this, we can infer the average ring number per primary aromatic cluster in individual *cha* cages to be around three to four. This inference accords with the preceding conclusions achieved by DFT calculations and MS analysis, further affording a substantive evidence toward the molecular structure of coke building unit and the cage-passing growth mechanism.

In doing so, this work, in large part, addresses the long-standing search for PAHs structure elucidation, and thus offers the mechanistic picture on molecule-resolved PAHs speciation and evolution. Thanks to the direct identification of the structure of the condensed PAHs, this work provides a full spectrum understanding of the molecular routes evolving from occluded long chain olefins to soluble lighter PAHs (ranging from naphthalene to pyrene), and eventually to insoluble heavier PAHs (cross-linked multicore aromatics that were established in this work) in the exemplary methanol conversion reaction as shown in Fig. 5.

These results also demonstrate MALDI FT-ICR MS (integrating with isotope labeling) working as a powerful analytic approach to elucidate the complex molecular structure of insoluble coke largely present in catalysis chemistry. Although being merely applied to probe the molecular structure of PAHs formed in cage-structured zeolites or molecular sieves in the current work, our developed PAHs analytic strategy is expected to be of more general validity in terms of the following discussions.

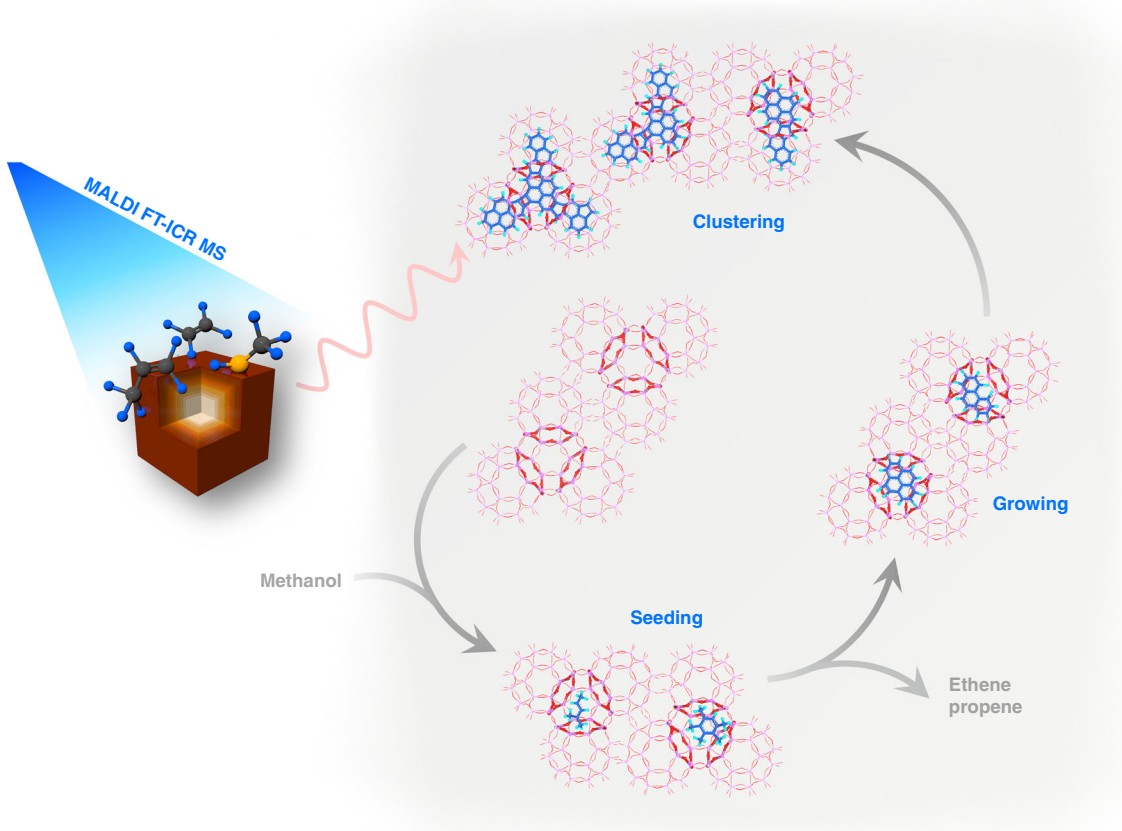

**Fig. 5 Molecular evolution route of PAHs.** PAHs evolve from occluded long chain olefins and light polymethylbenzenes (coined "seeding") to three- four-ring PAHs behaving as building units for aromatic cluster formation ("growing"), and eventually to cage-passing aromatic clusters ("clustering").

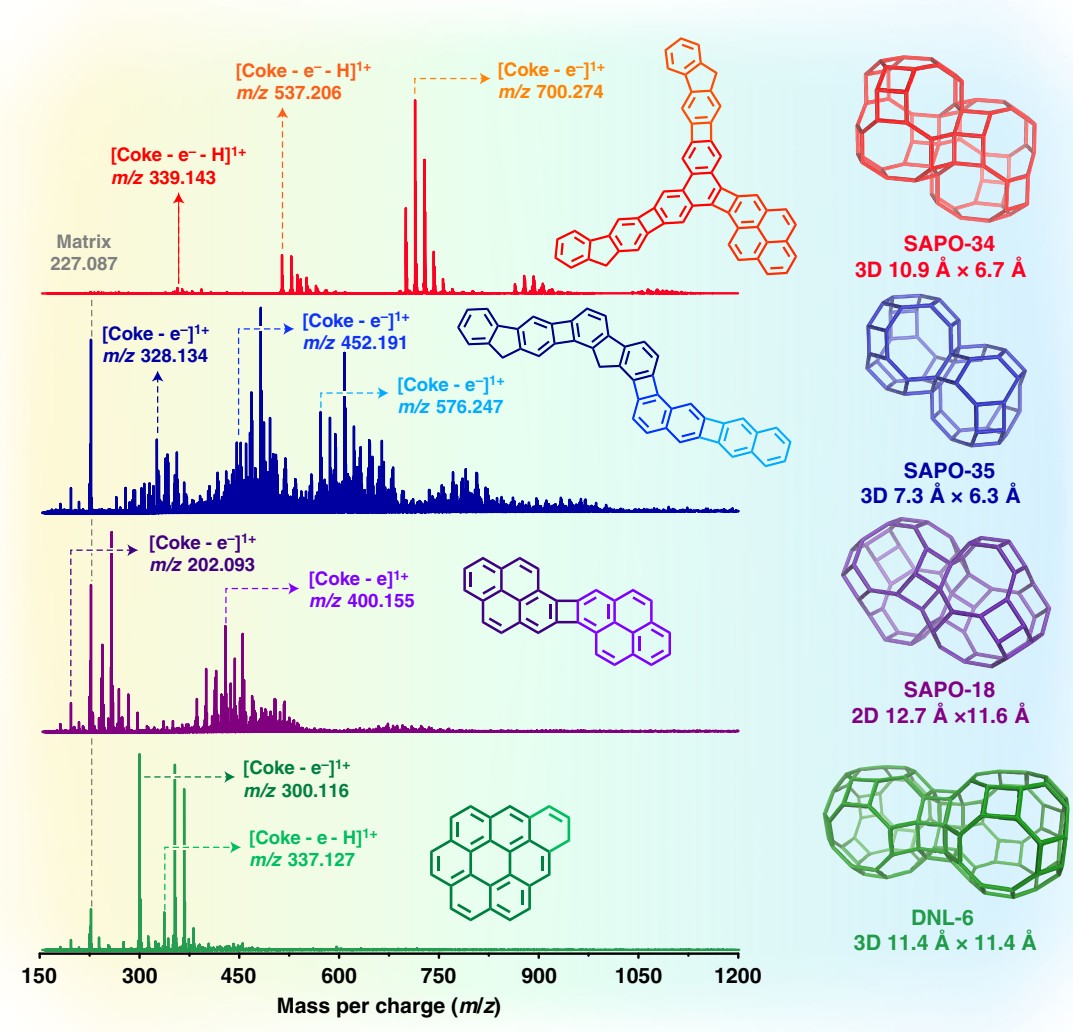

**Fig. 6 Molecular structure analysis of PAHs in other cage-structured molecular sieves.** MALDI FT-ICR mass spectra for extracts obtained from other cage-structured molecular sieves and the deduced possible molecular structure of PAHs. The indicated molecular structures are representative examples of possible structures.

Coupling of aromatic species via biphenyl to form more condensed PAHs was previously hypothesized to be present in channel-structured zeolites, such as HZSM-5[46] and HFER;[47] besides, aliphatically bridged multicore PAHs were also proposed in soot formation chemistry[1]. With the similar structural feature of PAHs, the means and methodology provided here are expected to be versatile for deciphering the formation and evolution of PAHs involved other catalysis systems and fields.

Next, to test the generality of cage-passing coking model, we performed the structural analysis of retained coke species on other types of cage-structured molecular sieves, such as SAPO-35 (**LEV**), SAPO-18 (**AEI**), DNL-6 (**RHO**), also taking methanol conversion as a model reaction. The physicochemical properties of these three catalysts are compiled in Supplementary Table 4 and Supplementary Fig. 12. The methanol conversion performances and the GC-MS analysis of the CCl$_4$-extracted soluble coke on three catalysts are shown in Supplementary Figs. 13 and 14. The coke molecular structures were determined on the basis of the masses of maximum peaks, given by MALDI FT-ICR MS

and complied in Fig. 6, along with the specific cage structures and dimensions. For SAPO-35 with small cage dimensions (7.3 Å × 6.3 Å), the deduced coke molecule is found to be directly coupled via biphenyl mode between the coke units of naphthalene and fluorene. The predicted coking units are consistent with the GC-MS analysis of the extracts after dissolution-extraction experiments, in which both naphthalene and fluorene were detected with the former as the main constituent (Supplementary Fig. 14). Pyrene was found to be the exclusive coke unit on SAPO-18, which is related to its large cage dimensions (12.7 Å × 11.6 Å), also according with the GC-MS results with pyrene as the dominating aromatic species (Supplementary Fig. 14). Nevertheless, the case was quite different for DNL-6 (11.4 Å × 11.4 Å), for which no cage-passing event occurred and coronene and its derivatives were detected as the dominant heavier coke, in excellent agreement with the GC-MS analysis with coronene being the main species (Supplementary Fig. 14). Such striking differences between DNL-6 and other cage-structured molecular sieves (SAPO-34, SAPO-35, and SAPO-18 used here) lies in the

distinct cage-connecting ways; *rho* cages are connected by double 8-MR while cages for other molecular sieves are connected by single 8-MR. The distance between the double 8-MR is around 180 pm, longer than the bond length of normal C–C (154 pm), thus preventing the cage-passing coupling of coke building unit. The results obtained on DNL-6, in turn, reflect the rationality of cage-passing growth on the cage-structured zeolites. These results, altogether, serve as a proof of concept for cage-passing growth mechanism.

As one kind of critical end-result of carbon resources, the speciation and evolution of these insoluble heavier PAHs represent the structure–performance relations in host–guest chemistry. The understanding of the subtle fingerprints of these heavier PAHs embodies the concept of cavity-control and the shape selective catalysis in zeolite- or molecular sieve-catalyzed reactions. Coking is a shape-selective process: the chemical nature of coke building units is cavity-controlled in terms of cavity structure and dimensions.

**A deactivating model with "coke" spatial distribution**. We have previously observed the nonuniform distribution of Si (that is, Si enriching in the near-surface zone of crystals) in SAPO-34[48] and other SAPO molecular sieves[49]. Such inhomogeneity of Si was even observed in low Si SAPO-34[10]. The segregation of Si concurrently results in higher acid density at the near-surface zone of crystals[16], eventually leading to the nonuniform spatial distribution of coke. Consequently, the bulky coke species (clusters) are inclined to be formed at the shell layers of SAPO-34 impeding the molecule diffusion[50]. This argument is conclusively supported by the direct observation of coke spatial distribution visualized by advanced CFM measurements[10,50] and APT analysis[16].

The above analysis, together with the exclusion of apparent external coke deposition and a full-spectrum molecular structure of "coke" identified by the multiple techniques presented in this work, allows us to formulate the deactivation mechanism and propose a compelling coking and deactivation model, at a single crystal scale, with the detailed information of chemical nature and spatial distribution of "coke" molecules (Supplementary Fig. 15). The cross-linked, multicore aromatic hydrocarbon clusters (with fluorene, phenanthrene and pyrene as building unit) are prone to reside at the shell layer of SAPO-34. Compared with the direct imaging of these bulky coke clusters by the state-of-the-art APT technique[16], our work advances the chemical understanding of the structural nature of these heavier PAHs. Such near-surface enriched, cross-linked bulky coke clusters would impede the diffusion of reactants, thereby frustrating the structural extension of the lighter aromatic species inside the near-core part of crystal, eventually making part of in-core acid sites unutilized[10,50].

In practical MTO industry, pre-coking technology for fresh catalyst[32], and partial regeneration technology for the deactivated catalyst are adopted to improve the lower olefin selectivity in the initial reaction stage. Our well-built deactivation model (Supplementary Fig. 15), especially for the chemical nature and spatial location of previously unrecognized cage-passing PAHs, would serve as a solid basis for mechanism exploration and process design of these two critical industrial technologies.

## Discussion

In this work, we have demonstrated, through coupling the state-of-the-art MALDI FT-ICR MS with isotope labeling technique, the cage-passing growth concept (forming cross-linked multicore aromatics) by identifying the structural fingerprints of PAHs in an exemplary, industrially important SAPO-34-catalyzed methanol conversion reaction. As a proof of concept, cage-passing growth mechanism was validated on other cage-structured zeolites, and is expected to be a general C–C assembly mode for PAHs growth in zeolite- or molecular sieve-catalyzed reactions. Such a mechanistic motif unveils the significance of host–guest chemistry in catalysis field and would help in resolving the long-standing scientific puzzle of how to definitely identify the chemical nature of PAHs. The cross-linked multicore molecular structure of PAHs identified in this work resembles the structure of petroleum asphaltene[51], coal[52], and soot formed in fuel combustion process[1]. These common features enable the unifying description of chemical structure motif of a large category of extended PAHs as biphenyl- and/or alkyl group-bridged, three dimensionally cross-linked multicore aromatics. These insights would help in establishing the molecular structural evolution of carbonaceous matter in much broader chemistry fields.

## Methods

**Catalyst**. All synthesis experiments of the molecule sieves used in this work were performed by the traditional hydrothermal method according to refs. [53,54]. SAPO-34, SAPO-18, SAPO-35, and DNL-6 were synthesized using triethylamine, N, N-diisopropylethylamine, hexamethyleneimine, and N-methylbutylamine as organic structure directing agents, respectively. A typical synthesis procedure is described as follows. Orthophosphoric acid, tetraethyl orthosilicate, aluminum isopropoxide, and amines were added sequentially into deionized water and then vigorously stirred overnight. Exclusively, for DNL-6, a certain amount of seeds (2–5 wt%, based on $Al_2O_3$) and hexadecyl trimethyl ammonium bromide (CTAB) were added to the homogeneous gel. The mixed gel was transferred into a Teflon-lined stainless-steel autoclave and crystallized at a determined temperature under autogenously pressure. The final solid products were recovered by filtration, washed with deionized water repeatedly, and dried at 110 °C overnight. The gel chemical compositions of SAPO-34, SAPO-18, SAPO-35, and DNL-6 were 1.0 $Al_2O_3$: 1.0 $P_2O_5$: 0.6 $SiO_2$: 3 $NEt_3$: 50 $H_2O$, 1.0 $Al_2O_3$: 1.0 $P_2O_5$: 0.6 $SiO_2$: 1.6 $C_8H_{19}N$: 50 $H_2O$, 1.0 $Al_2O_3$: 1.0 $P_2O_5$: 1.0 $SiO_2$: 1.5 $C_6H_{13}N$: 50 $H_2O$ and 1.0 $Al_2O_3$: 0.8 $P_2O_5$: 0.6 $SiO_2$: 4 $C_5H_{13}N$: 100 $H_2O$: 0.4 CTAB (20% seed), respectively. The crystallization conditions were 200 °C for 48 h (SAPO-34), 170 °C for 48 h (SAPO-18), 200 °C for 24 h (SAPO-35), and 200 °C for 24 h (DNL-6). The H-form SAPO-34, SAPO-18 and DNL-6 were obtained by calcining the crystallized products at 550 °C in air for 4 h to remove organic templates, and H-SAPO-35 was obtained by calcination at 700 °C in air for 4 h before use. The heating rate for both calcination procedures was 1.5 °C min$^{-1}$ from ambient temperature to target ones.

**Characterization**. The morphology of catalysts and the coke isolated from the spent catalyst by filtration of the extracts was observed by scanning electron microscope using a Hitachi TM3000 with an accelerating voltage of 15 kV.

Powder XRD of catalysts was recorded on PANalytical X'Pert PRO X-ray diffractometer equipped with Cu Kα radiation ($\lambda = 0.15418$ nm) as the X-ray source, operating at 40 kV and 40 mA. XRD patterns were recorded from 5° to 60° with a scan speed of $2\theta = 5.0°$ min$^{-1}$.

The chemical compositions of catalysts were determined with a Philips Magix-601 X-ray fluorescence spectrometer.

X-ray photoelectron spectra were recorded using a Thermo ESCALAB 250Xi instrument equipped monochromatized Al Kα (1486.6 eV, 15 kV, 10.8 mA) X-ray. Sample powders were pressed into pellets with ca. 5 mm diameter and a few tenth of millimetre thickness. The binding energy values were referenced to the C *1 s* line of residual carbon at 284.8 eV. The atomic ratio was calculated using the peak areas of Si *2p*, Al *2p*, and P *2p*, respectively.

The surface area and pore volume for the fresh and spent catalysts were determined by $N_2$ adsorption-desorption isotherms performed at liquid $N_2$ temperature on a Micromeritics ASAP 2020 analyzer. Before $N_2$ sorption, the fresh catalyst was outgassed at 350 °C in vacuum for 4 h, and the spent catalysts were outgassed at 90 °C in vacuum for 1 h followed by 3 h at 250 °C in order to remove the adsorbed water totally and at the same time, to avoid the coke decomposition[55]. The specific surface area was calculated based on the Brunauer–Emmett–Teller equation taking the adsorption data in the range of $p/p_0 = 0.05$–0.2. the external surface area, micropore area and micropore volume were determined by the t-plot method taking the adsorption data at $p/p_0 = 0.2$. The total pore volume was determined by the $N_2$ adsorption amount at $p/p_0 = 0.972$.

The "coke" amounts of the spent catalysts were measured by thermogravimetric analysis (TGA) analysis performed on a SDTQ 600 analyzer with a temperature-programmed rate of 10 °C min$^{-1}$ under flowing air (100 ml min$^{-1}$) from room temperature to 900 °C. The weight loss and the accompanying endothermic peaks below 250 °C are ascribed to the desorption of water (Supplementary Fig. 16).

**Liberating carbonaceous deposits from spent catalysts**. The retained species were liberated by a modified HF dissolution-solvent extraction method. Typically, 50 mg of the spent catalysts were firstly dissolved in 0.5 ml 20% HF solution in a Teflon vial. Upon the total digestion of the zeolite framework and the evaporation

of unreacted HF in fume hood, the released carbonaceous species were extracted by 0.5 ml tetrachloromethane ($CCl_4$). Compared with the classical Guisnet method[4], two improvements were made in our modified dissolution-extraction experiments: (1) No neutralization of HF was taken, aiming to avoid massive heat release that will result in violent boiling in the Teflon vial and to avoid forming $AlF_3$ precipitate that will impede the isolation and analysis of insoluble coke; (2) $CCl_4$ instead of methylene chloride ($CH_2Cl_2$) was adopted here as extractant, in that $CCl_4$ with density of $1.59 \text{ g cm}^{-1}$ is heavier than $CH_2Cl_2$ ($1.33 \text{ g cm}^{-1}$), and the solubility of $CCl_4$ in water ($0.081 \text{ g } 100 \text{ mL}^{-1}$) is much lower than that of $CH_2Cl_2$ ($17.5 \text{ g } 100 \text{ mL}^{-1}$)[56], thereby enabling the layering of solution more obvious and more carbonaceous species being recovered.

**Isolating insoluble coke**. After HF dissolution-$CCl_4$ extraction, the insoluble fraction of coke (suspended in the oil phase or appearing as black particles floating between aqueous phase and oil phase) was recovered by suction filtration, followed by careful purification with $CCl_4$ to remove the absorbed organic species.

**Structural identification of carbonaceous deposits**. The chemical nature of the extracted carbonaceous deposits was complementarily identified by GC-MS and MALDI FT-ICR MS.

The GC-MS analysis was performed on Agilent 7890 A Gas Chromatograph equipped with a HP-5 capillary column and a FID detector, with hexachloroethane ($C_2Cl_6$) added as internal standard to determine the light soluble "coke" amount. Prior to measurement, a calibration of GC-MS was performed for precisely quantifying the soluble "coke". Firstly, 50.5 mg of toluene, 50.3 mg of dimethyl-naphthalene, and 50.4 mg of pyrene were added in sequence into 25 mL of $CCl_4$ with $C_2Cl_6$ as internal standard. Then, the solution was diluted for four times with $CCl_4$ to obtain a series of solution with different solute concentrations. The calibration line was shown in Supplementary Fig. 17.

Before measurements, series of control experiments, using matrix (dithranol), model compound of 9,10-di-(2-naphthyl) anthracene and extracted coke species from deactivated catalysts as standard or reference samples (Supplementary Fig. 7), were conducted to determine the optimal laser output to be 18%, under which the samples would not suffer from polymerization or cleavage.

For MALDI FT-ICR MS measurements of carbonaceous deposit, the samples were prepared and analyzed in three ways. In the first mode, the deactivated samples were directly measured with MALDI FT-ICR MS, attempting to extract the structural information of the potential coke species located on the outer surface of the catalyst[17,24]. Specifically, a solution of $5 \text{ mg mL}^{-1}$ dithranol (MALDI matrix) in tetrahydrofuran (THF) was prepared, and 5 mg of the deactivated catalysts were suspended in 500 μL of this solution. 1 μL of the mixture was spotted onto the sample holder. After air-drying at room temperature, MS signals were recorded in positive ion mode on a 15-T FT-ICR mass spectrometer (Solarix XR, Bruker Daltonics, Bremen, Germany) equipped with a Nd: YAG laser emitting 355 nm laser to generate ions. Ion source parameters were optimized to a broadband range ($150 < m/z < 1200$) for coke species analysis. Ionization was achieved with 200 Hz and 32 shots were summated in random handed walk with a grid width of 1000 μm. The mass spectrometer was calibrated before measurement by a peptide calibration standard.

In the second mode, the extracts were measured on the same instrument using the same setting as deactivated catalyst measurements. 10 μL of the extracts were mixed with 10 μL of THF containing dithranol as MALDI matrix ($5 \text{ mg mL}^{-1}$ in THF). After sonication, 1 μL of mixture was spotted onto the sample holder and dried at room temperature. Then, the sample holder was introduced into the ion source of the mass spectrometer for MS analysis.

In the third mode, the isolated insoluble coke was measured as the aforementioned procedure. 1 mg of the collected insoluble coke was suspended in 200 μL of THF with dithranol as MALDI matrix ($5 \text{ mg mL}^{-1}$ in THF).

**Solid state MAS NMR**. The $^{13}C$ magic-angle spinning nuclear magnetic resonance (MAS NMR) experiment was performed on a Bruker Avance III 400 spectrometer equipped with a 9.4 T wide-bore magnet using a 0.7 mm H-X Ultra-High-Spin MAS probe. The spectrum was recorded using high-power proton decoupling sequence at a spin rate of 110 kHz with a π/4 pulse width of 1.9 μs and a 60 s recycle delay. Chemical shift was referred to adamantane at 28.4 ppm.

The solid-state $^1H$ and $^{29}Si$ MAS NMR experiments were performed on a Bruker Avance III 600 spectrometer equipped with a 14.1 T wide-bore magnet using a 4 mm H-X MAS probe. $^1H$ MAS NMR spectrum was recorded with a π/4 pulse of 2.0 μs and 10 s recycle delay using a single pulse sequence at 12 kHz. Chemical shift was referred to adamantane at 1.74 ppm. Before NMR test, the sample was dehydrated typically at 420 °C under vacuum. $^{29}Si$ MAS NMR spectrum was recorded using high-power proton decoupling sequence at 12 kHz with a π/4 pulse width of 2.8 μs and a 10 s recycle delay. Chemical shift was referred to 4,4-dimethyl-4-silapentane sulfonate sodium salt at 0 ppm.

**Computational details**. For theoretical calculations of stability of individual aromatic species adsorbed in one single cage. All the DFT calculations were performed using the plane-wave pseudopotential method, implemented with the Dmol3 code. The Perdew–Burke–Ernzerhof (PBE) exchange-correlation functional was used to describe the exchange-correlation effects. Interaction between the valence electrons and the ion core was substituted by an ultrasoft pseudopotential. The self-consistent convergence accuracy was set at $10^{-5}$ Hartree per atom, and the maximum displacement was $5 \times 10^{-3}$ Å. Here, a single cell was employed to simulate SAPO-34 systems. A Monkhorst-Pack of k-point and $1 \times 1 \times 1$ was used to sample the Brillouin zone for geometry optimization and for calculating the energy of the SAPO-34 systems including aromatics.

Periodic DFT calculation has been performed to investigate the energy for CHA-typed AlPO-34 and coke. The initial structure of pure siliceous zeolite was taken from the International Zeolite Associations database[57], subsequently Al and P atoms were substituted, and finally the structure was optimized. Then $2 \times 2 \times 1$ super cells were used to study the interaction energy. The optimization was performed in the mixed Gaussian plane wave scheme using the CP2K code (https://www.cp2k.org/)[61,62]. The PBE exchange-correlation functional[58] was applied and the D3 correction[59] of Grimme was used to account for the dispersion interactions. During the optimization, the structures were relaxed by using the DZVP basis set and GTH pseudo potentials[60]. The plane wave cutoff energy and relative cutoff was 650 Ry and 60 Ry, respectively.

**Methanol conversion reaction test**. Methanol conversion reactions were performed on a fixed-bed quartz reactor with an internal diameter of 8 mm under atmospheric pressure. The catalyst was pressed, crushed and sieved to 250–420 μm and then 600 mg of catalysts equally separated into four layers by quartz wool were packed into reactor. Prior to reaction, the catalysts were activated at 480 °C in $N_2$ for 1 h, and then the temperature was adjusted to the set value. Afterwards, saturated methanol vapor was fed into the reactor by passing the carrier gas ($N_2$) through a saturator kept at 25 °C. The effluent products were analyzed by online gas chromatograph (Agilent GC 7890 A) equipped with capillary column HP-Plot/Q-HT (30 m × 0.32 mm × 20 μm) and a FID detector. The temperature of the transferring line was maintained at 190 °C to avoid the condensation of effluent products. The conversion and selectivity for MTO reaction were calculated on $CH_2$ basis. Dimethyl ether was considered as reactant due to the fast equilibrium between methanol and dimethyl ether.

## Data availability

All data supporting the findings of this study are available within the article and its Supplementary Information, and/or from the corresponding authors upon reasonable request.

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

## Acknowledgements

This paper is dedicated to the 70th anniversary of the Dalian Institute of Chemical Physics, Chinese Academy of Sciences. The authors thank the financial support from the National Natural Science Foundation of China (Grant No. 21703239, 91745109, 21972142 and 21991092), the Key Research Program of Frontier Sciences, Chinese Academy of Sciences (QYZDY-SSW-JSC024), and International Partnership Program of Chinese Academy of Sciences (121421KYSB20180007). The authors wish to acknowledge Prof. Renan Wu and Dr Lihong Wan in Dalian Institute of Chemical Physics, Chinese Academy of Sciences and Dr Mingjian Luo in Northeast Petroleum University for the fruitful discussions in MALDI FT-ICR mass spectra analysis.

## Author contributions

N.W. and Y.Z. conceived, coordinated the research, and designed the experiments. The two corresponding authors Zho.L. and Y.W supervised the project and led the collaboration

efforts. The MTO reaction, MALDI FT-ICR MS measurements combined with isotope labeling as well as other characterizations were performed by N.W. and Y.Z. and later they finished the data treatments and analyses. Zhi.L., A.Z., W.Z., and J.H. performed DFT calculations. T.S. and S.X. performed solid-state NMR characterization. S.L. provided the used catalysts with different deactivation extents. Y.H. contributed to the general characterization of spent catalysts. All authors contributed to analysis and discussion on the data.

## Competing interests

The authors declare no competing interests.
