## [Peer Review File · Nature Communications]

Reviewers' comments:

Reviewer #1 (Remarks to the Author):

This manuscript by Wang et al show new information on the structure of deactivating coke in the commercial methanol to olefins (MTO) catalyst SAPO-34. As well stated in the introduction, this reaction is of high industrial importance and research on the complex chemistry involved in its mechanism helps to understand and improve zeolite properties for this and other applications in catalytic hydrocarbon interconversions. The scientific results are well described, systematically obtained by well-designed experiments and without oversimplifications. The conclusions reached are supported by abundant evidence. The novelty of the work is clear alone for SAPO-34, but the fact that the authors have extended the study to other cage-containing zeolites with different sizes and interconnectivity, as shown in the last part, makes it highly interesting. For all these, I think the manuscript is suitable for publication in Nature Communications, and I only have minor comments for the authors:

1. The introduction is nicely written but I am missing the motivation of the study (later explained in the discussion), which I think is key for the broad readership of NatComm. It should be emphasized at the beginning why is it important to have a full molecular description of the coke deactivating the zeolite micropores (and not only the location).
2. The information in some Figures is very crowded and stemming from different techniques. For instance Fig 2 almost look like a crowded graphical abstract. I would suggest taking instead Fig 2e to SI, as it is the least related one. After all, the information about DFT calculation of the PAH structures and energies can be conveyed by the text of the manuscript (as it is actually in lines 6-10 in page 9).
3. Similar problem with Figure 3. In this case, the footnotes on Figure 3e table are not provided in the caption. Also using same letters a to e for footnotes of the table as for the Figure panels would be confusing for the reader. The numeric values shown in the Table are not essential for understanding the manuscript, although they are interesting for scientists in the field and should be of course included in SI. The molecular structures proposed for PAH appearing in Fig 3e could be also left out: these molecules are already shown in Figure 4, where other data regarding assignment of MS spectra are displayed.
4. It is not very clear how solid state ^{13}C NMR supports the structures proposed. For instance, it is mentioned that the aromatic bridgehead vcarbon value gives the size of the aromatic cluster, but can one see information about the interconnectivity of these clusters? Can these C-NMR results be interpreted in a different way, for instance as isolated 3- and 4-ring PAH trapped in the CHA cages? It would be useful to provide expected percentages of the different parameters in Figure S11 for candidate molecules.

5. There is a typo in reference 9, it is year 2008, not 2010.

Reviewer #2 (Remarks to the Author):

The manuscript of Nan Wang et al. reports a study of coke molecules in the cages of mainly SAPO-34. They found a way to identify the coke clusters using MALDI FT-ICR and isotope studies. The authors discovered with experiments that there is no coke on the external surface of SAPO-34, in contrast to BEA and ZSM-5, which are deactivating from the outside. Their main finding is that they found cage passing growth mechanism for the coke molecules in zeolites, which contain cages. With the combination of MALDI FT-ICR and isotope studies they came close to the actual determination of the identity of the cage passing coke molecules. In my opinion the authors could decrease the value of some of their statements (see comments below). Nevertheless, this manuscript contains a novel combination of techniques and gives, besides that, a good overview about the state-of-art of the coking behaviour of zeolites in the MTO process.

In conclusion, I think this manuscript could be suitable for publication in Nature Communications with some minor as well as major revisions. My comments are given below.

Main comments

Introduction

- A link is made with the whole chemistry of PAHs from interstellar space to catalysis chemistry. However, it seems this link is made to enhance the importance of the work and no valuable link between the actual work and unrevealing of the whole PAHs chemistry is made. Improvement of the explanation of the relationship is necessary or the statements have to be weakened. This comment is also applicable for the abstract.

Results

- Separating the catalyst bed in 4 layers, while using 600 mg of catalyst, will not completely solve the problem of inhomogeneous axial deactivation. Even in the layers this phenomenon will appear and

therefore it cannot be stated that by just using the upper layer this problem is solved. Reformulation is necessary.

- The micropore volume is determined with N₂-physisorption. When pores are blocked on the outer shell of the zeolite the N₂ diffusion is also hindered and this could lead to an incorrect correlation between the coke content and the pore volume as the authors describe. There is no explanation in figure 4 of the supplementary information about the way the authors got to these coke contents. This is a necessary requirement as than it would be clear it is about a fully deactivated catalysts or a partially coked catalyst. Figure 4 of the supplementary information is thereby also going to 16 wt% coke, while the results of figure 2c show coke contents up to 25 wt%. Elaboration and clarification is these data is necessary.

- In my opinion, the strongest argument for no formation of PAHs on the outer surface is the MALDI FT-ICR MS experiments of the authors itself (where they show that when the zeolite is not dissolved, there are no remarkable carbonaceous species on the external surface of SAPO-34, in contrast to found data in literature about ZSM-5 and BEA). I would therefore recommend combining all these arguments together in one paragraph rather than mentioning the first three argument and the experimental arguments in different paragraphs.

- More elaboration on the results of the DFT calculations for the cage passing principle are necessary.

Conclusion

- Such a mechanistic ... catalysis field. Too strong conclusion, the authors did not solve the problem completely, but definitely helped to solve the "puzzle" in the future.

Minor comments

Abstract

- Second sentence is too long and is not correct English. Dividing the sentence in multiple sentences would increase the readability of the text.

- Mention how the "mechanistic concept is theoretically and experimentally substantiated".

Introduction

- In my opinion, syngas transformation has be changed into syngas conversion

- Additionally, the formation of PAHs in zeolite confined spaces is not only determined by the size, shape en morphology of the zeolites but by many other factors. It seems like the authors mean that they choose the MTO mechanism because the growth is determined by the confined space. Reformulation is necessary.

Results

- Statement about the accumulation of coke being beneficial for the ethene selectivity: Yes that is true, as the mechanism is believed to be going through a hydrocarbon pool. However, the authors state that the PAHs are also beneficial. This is proved not to be the case in many other studies as showed with operando spectroscopy.

- The lattice expansion in the c-direction has been addressed to the formation of coke insight the cages of the zeolite in previous work (J. Goetze et al. 2018). The authors “reasonably postulate” that the expansion will not have an effect on the accommodation of the aromatic molecules. In order to be fully correct, this sentence has to be reformulated.

- The first argument for why the authors exclude the possibility of external coke on SAPO-34. They state that there should be non-selective reactions when there are acid sites on the outside. This requires a reference.

Methods

- Heating ramp of calcination is missing

References

Should be more up to date; with also some acknowledgment of other work on this topic of carbon deposits; and hydrocarbon identification.

Graphical Abstract

The graphical abstract can be ecstatically improved. Suggestion would be to use Adobe colour wheel.

Additionally, it appears that the main conclusion of this paper is that the coke molecules are decreasing in size towards the centre of the crystal. However, this is not the conclusion of the work of the authors but a conclusion made within other publications before. I understand why they want to represent it this way, but this is not what they actually discovered. They discovered the cage crossing principle and therefore, in my opinion, the graphical abstract should focus on that.

Response to the reviewer's comments

We thank all the reviewers for the detailed remarks on our manuscript. All reviewers' feedbacks have been carefully considered and we have revised the manuscript based on these suggestions.

This document summarizes our major changes to the manuscript, as well as a point-by-point discussion of the reviewer's remark. The reviewer's comments are reproduced verbatim below, along with our responses (in blue). We also have enclosed a revised version of the paper in which the changes are highlighted in red.

Reviewer #1 (Remarks to the Authors):

This manuscript by Wang et al show new information on the structure of deactivating coke in the commercial methanol to olefins (MTO) catalyst SAPO-34. As well stated in the introduction, this reaction is of high industrial importance and research on the complex chemistry involved in its mechanism helps to understand and improve zeolite properties for this and other applications in catalytic hydrocarbon interconversions. The scientific results are well described, systematically obtained by well-designed experiments and without oversimplifications. The conclusions reached are supported by abundant evidence. The novelty of the work is clear alone for SAPO-34, but the fact that the authors have extended the study to other cage-containing zeolites with different sizes and interconnectivity, as shown in the last part, makes it highly interesting. For all these, I think the manuscript is suitable for publication in Nature Communications, and I only have minor comments for the authors.

Our Response: We sincerely thank the reviewer for the positive assessment and comments on our work.

1. The introduction is nicely written but I am missing the motivation of the study (later explained in the discussion), which I think is key for the broad readership of NatComm. It should be emphasized at the beginning why is it important to have a full molecular description of the coke deactivating the zeolite micropores (and not only the location).

Our Response: Following the reviewer's instructive advice, we have added the motivation of the study associated with the reason for a full molecular description of the coke deactivating the zeolite in the revised manuscript. The corresponding sentences in p.3 has been revised to *"To alleviate or even prevent catalyst deactivation, unambiguously unravelling of the structural motifs of PAHs, at a molecular scale, is necessary. Despite enormous endeavors, a full-molecular description of the precise chemical structures of these PAHs is far from being achieved. Yet, if a full-spectrum deciphering of such PAHs is realized, it would become feasible to trace and uncover*

the molecular route of PAHs formation and evolution, and thus to obtain unprecedentedly detailed structural information of coke deactivating the zeolite. This would, in turn, help in improving industrial processes. Hence, this work aims to definitely fingerprinting the molecular identity of those complex PAHs.”

2. The information in some Figures is very crowded and stemming from different techniques. For instance Fig 2 almost look like a crowded graphical abstract. I would suggest taking instead Fig 2e to SI, as it is the least related one. After all, the information about DFT calculation of the PAH structures and energies can be conveyed by the text of the manuscript (as it is actually in lines 6-10 in page 9).

Our Response: We appreciate the reviewer’s kind suggestion. As suggested, we have already put Fig. 2e into Supplementary Information as Supplementary Fig. 4 in the revised manuscript so that Fig. 2 would look less crowded and more readable.

3. Similar problem with Figure 3. In this case, the footnotes on Figure 3e table are not provided in the caption. Also using same letters a to e for footnotes of the table as for the Figure panels would be confusing for the reader. The numeric values shown in the Table are not essential for understanding the manuscript, although they are interesting for scientists in the field and should be of course included in SI. The molecular structures proposed for PAH appearing in Fig 3e could be also left out: these molecules are already shown in Figure 4, where other data regarding assignment of MS spectra are displayed.

Our Response: We appreciate the reviewer’s carefulness and kind suggestion. We are sorry for the missing footnotes on Fig. 3e table in the original manuscript, and that have been added in the revised manuscript. Also, we have moved this Fig. 3e table into Supplementary Information as Supplementary Table 3. Indeed, the molecular structures proposed for PAHs are already displayed in Fig. 4, so for simplification the molecular structures in Fig. 3e (that is Supplementary Table 3 in the revised manuscript) has been left out.

4. It is not very clear how solid state ^{13}C NMR supports the structures proposed. For instance, it is mentioned that the aromatic bridgehead carbon value gives the size of the aromatic cluster, but can one see information about the interconnectivity of these clusters? Can these C-NMR results be interpreted in a different way, for instance as isolated 3- and 4-ring PAHs trapped in the CHA cages? It would be useful to provide expected percentages of the different parameters in Figure S11 for candidate molecules.

Our Response: The solid state ^{13}C NMR first demonstrated the aromatic nature of the majority of carbon atoms in the coke with less aliphatic carbon. More importantly, it gives the size of per primary aromatic cluster in individual *cha* cages to be 3- to 4-ring

PAHs by the aromatic bridgehead carbon value. This conclusion further supports the molecular structure of coke building unit that deduced by DFT calculations and MALDI FI-ICR MS analysis, as discussed in the manuscript. Unfortunately, we cannot see information about the interconnectivity of these clusters, although an advanced solid state NMR with ultra-high rotation rate up to 1.1 k s^{-1} has been applied. This may be due to the intrinsic resolution limitation of NMR technique. And according to our knowledge, there is no reported spectroscopic signal given by NMR technique that can afford information about the interconnectivity of aromatic structural unit. But, as discussed in the manuscript our developed MALDI FT-ICR MS integrated with isotope labelling technique has clearly demonstrated the cage-passing interconnectivity of these aromatic clusters. Even via the systematic identification and screening process, two possible structures are proposed as shown in Fig. 4. Unfortunately, the percentage of them cannot be determined by either MALDI FT-ICR MS or DFT calculation.

5. There is a typo in reference 9, it is year 2008, not 2010.

Our Response: We appreciate the reviewer's carefulness and have fixed this typo. The year has been corrected to 2008 in reference 10 in the revised manuscript.

Reviewer #2 (Remarks to the Author):

The manuscript of Nan Wang et al. reports a study of coke molecules in the cages of mainly SAPO-34. They found a way to identify the coke clusters using MALDI FT-ICR and isotope studies. The authors discovered with experiments that there is no coke on the external surface of SAPO-34, in contrast to BEA and ZSM-5, which are deactivating from the outside. Their main finding is that they found cage passing growth mechanism for the coke molecules in zeolites, which contain cages. With the combination of MALDI FT-ICR and isotope studies they came close to the actual determination of the identity of the cage passing coke molecules. In my opinion the authors could decrease the value of some of their statements (see comments below). Nevertheless, this manuscript contains a novel combination of techniques and gives, besides that, a good overview about the state-of-art of the coking behaviour of zeolites in the MTO process.

In conclusion, I think this manuscript could be suitable for publication in Nature Communications with some minor as well as major revisions. My comments are given below.

Our Response: We sincerely appreciate the reviewer's favorable comments and constructive suggestions on our work. All the reviewer's following suggestions have been carefully considered and we have done our best to address the reviewer's concerns to further improve our work.

Main comments

Introduction

- A link is made with the whole chemistry of PAHs from interstellar space to catalysis chemistry. However, it seems this link is made to enhance the importance of the work and no valuable link between the actual work and unrevealing of the whole PAHs chemistry is made. Improvement of the explanation of the relationship is necessary or the statements have to be weakened. This comment is also applicable for the abstract.

Our Response: According to the reviewer's kind suggestion, we have weakened and improved these statements in the Abstract and Introduction sections of the revised manuscript.

The sentence "Extension and clustering of polycyclic aromatic hydrocarbons (PAHs) are key mechanistic steps for carbonaceous matter formation in interstellar spaces and combustion processes, as well as for coking and deactivation in catalysis reactions." in the Abstract was revised to "*Extension and clustering of polycyclic aromatic hydrocarbons (PAHs) are key mechanistic steps for coking and deactivation in catalysis reactions.*"

Also, the statement in the first paragraph of the Introduction was revised to "*PAHs are ubiquitous, ranging from combustion chemistry to catalysis chemistry. The chemistry of PAHs growth remains a long-pursued yet largely unresolved scientific puzzle, such as in the fields of incomplete combustion of fuels as well as hydrocarbon and biomass hydrolysis (wherein human body- and environment-detrimental soot are formed)¹⁻³. The PAHs growth in zeolite-catalyzed chemical reactions would be analogous to soot formation, as both processes refer to the extension and clustering of preliminarily formed aromatic molecules. But, the PAHs growth in catalysis process may display distinct features in that nanosized pores of zeolites would regulate the dimensions and structural motifs of PAHs.*"

Results

- Separating the catalyst bed in 4 layers, while using 600 mg of catalyst, will not completely solve the problem of inhomogeneous axial deactivation. Even in the layers this phenomenon will appear and therefore it cannot be stated that by just using the upper layer this problem is solved. Reformulation is necessary.

Our Response: Indeed, the inhomogeneous axial deactivation along the catalyst bed, like a burnt cigar, is intrinsic and inevitable in fixed-bed reactor. We agree that no matter how thin each catalyst layer is, the inhomogeneity of coking would not absolutely be eliminated. In order to reduce the influence of this coke spatiotemporal inhomogeneity to the minimum extent, we separated the catalyst bed into 4 thin layers (about 0.3 cm per layer) using quartz wool (Supplementary Fig. 1). The layer-by-layer analysis method was also adopted in other literature¹. Moderately, we have modified the sentence in p.6 "To avoid this problem, when performing MTO reactions we

separated the catalyst bed equally into four layers using quartz wool (Supplementary Fig. 1) as described in the Methods section.” to “*To reduce the effect of inhomogeneous axial deactivation within control, when performing MTO reactions we separated the catalyst bed equally into four layers using quartz wool (Supplementary Fig. 1) as described in the Methods section.*”

1. Wang, C. et al. Impact of temporal and spatial distribution of hydrocarbon pool on methanol conversion over H-ZSM-5. *J. Catal.* **354**, 138-151 (2017).

- The micropore volume is determined with N₂-physisorption. When pores are blocked on the outer shell of the zeolite the N₂ diffusion is also hindered and this could lead to an incorrect correlation between the coke content and the pore volume as the authors describe. There is no explanation in figure 4 of the supplementary information about the way the authors got to these coke contents. This is a necessary requirement as than it would be clear it is about a fully deactivated catalysts or a partially coked catalyst. Figure 4 of the supplementary information is thereby also going to 16 wt% coke, while the results of figure 2c show coke contents up to 25 wt%. Elaboration and clarification is these data is necessary.

Our Response: We sincerely thank the reviewer for the insightful feedback. From the careful analysis of the MALDI FT-ICR MS results as discussed in the manuscript, the bulky coke species of cage-passing PAH clusters that are mainly localized at the outer shell of the zeolite crystallite, occupy up to four neighboring cages after MTO reaction for 130 min with almost no methanol conversion (even though more larger PAHs clusters would also be formed). This conclusion suggests that the pores in the outer shell of the zeolite should not be totally blocked, but rather be partially blocked. This deduction is directly supported by the measured linear correlation between micropore volume and “coke” contents as shown in Supplementary Fig. 5 in the revised manuscript. Moreover, if the outer shell of the zeolite crystallite is completely blocked (forming graphite-like carbon deposits on the outer surface or outer shell of the zeolite, like the case of ZSM-5 deactivation), N₂ molecular would not diffuse into the inner pores and the correlation between micropore volume and “coke” content would appear a sharp decline but not a linear reducing trend.

The coke contents shown in Supplementary Fig. 5 were determined by TGA measurement with a formula: coke content (wt%) = (weight measured at 250 °C - weight measured at 900 °C) /weight measured at 900 °C. The detailed method was introduced in Method-Characterization section and data graph could refer to Supplementary Fig. 16 in the revised manuscript.

We are sorry for not clearly indicating the line data to the corresponding coordinate axes. The line showing the data of 25% does not indicate the coke content assigned to right Y axis but indicate the weight of insoluble coke assigned to left Y axis in Fig. 2c. In the revised manuscript, we have added the illustrative arrows in Fig. 2c to clearly guide the readers. It should be noted that the coke content in Fig.2c goes to 18% after 130 min reaction, and the coke content in Supplementary Fig. 5 goes to 16% after 80 min reaction owing to the missing of pore volume data for this sample.

- In my opinion, the strongest argument for no formation of PAHs on the outer surface is the MALDI FT-ICR MS experiments of the authors itself (where they show that when the zeolite is not dissolved, there are no remarkable carbonaceous species on the external surface of SAPO-34, in contrast to found data in literature about ZSM-5 and BEA). I would therefore recommend combining all these arguments together in one paragraph rather than mentioning the first three argument and the experimental arguments in different paragraphs.

Our Response: We appreciate the reviewer's kind suggestion. We have rearranged all the arguments and combined them together in one paragraph in p.9 in the revised manuscript. It reads: "*4) More solid direct evidence for the rather less extent of on-surface coking was delivered by the advanced MALDI FT-ICR MS. The deactivated SAPO-34 samples (without dissolving the framework by HF) were directly tested by MALDI FT-ICR MS to probe the possibility of carbonaceous residues deposited on the external surface. The similar MS method was applied on HZSM-5¹⁷ and HBEA²³ and the results showed massive coke molecules detected on the external surface. But, the same is not true for SAPO-34, as the MS results demonstrated no remarkable MS peak signals of carbonaceous deposits occurring on the external surface of SAPO-34 with different reaction or deactivation extents (Supplementary Fig. 6).*"

- More elaboration on the results of the DFT calculations for the cage passing principle are necessary.

Our Response: In the revised manuscript, the structural optimization of cage-passing PAHs with four possible conformations was further performed by the periodic density functional theory (DFT) calculation at PBE/DZVP level with the relaxing of the coke fragments and *cha* cages. The energetic estimations show that the strain energy of zeolite framework for Model 1a is extremely high ($105.88 \text{ kcal mol}^{-1}$) (Supplementary Fig. 9) and the framework deforms severely, which indicates that the cage-passing configuration of Model 1a is not so reasonable. While for Model 1b being structurally similar with Model 1a with 120 degree angles between each building block, the strain of the coke species and zeolite framework is significantly reduced to $30\text{-}50 \text{ kcal mol}^{-1}$, and adsorption energy is $20.79 \text{ kcal mol}^{-1}$. Considering the high temperature (up to $475 \text{ }^\circ\text{C}$) for the coke formation, such deformation could possibly be acceptable.

As for Model 2a of cage-passing PAHs with 3-D configuration, the deformation of the coke species is relatively high ($E_{\text{strain-coke}} = 74.92 \text{ kcal mol}^{-1}$). While for Model 2b with plain configuration (with 120 degree angles between each building block), the deformation of coke species is much lower with the energy of $26.33 \text{ kcal mol}^{-1}$, and the adsorption energy is $-39.8 \text{ kcal mol}^{-1}$ much stronger than that of Model 2a with the energy of $-7.12 \text{ kcal mol}^{-1}$.

Consequently, for both Model 1 and 2, the plain configurations of cage-passing PAHs (that is, Model 1b and Model 2b) are more energetically feasible. Accordingly,

we adjusted the 3-D configuration in the original manuscript to the plain configuration in the revised manuscript. Due to the complexity and the bulky dimensions of cage-passing PAHs molecules enabling it challenging for calculations, the calculations of formation energy for cage-passing steps are still going on and will be reported in our coming other work.

Supplementary Fig. 9 The configurational optimization for our proposed Model 1 and 2 for cage-passing PAHs.

Conclusion

- Such a mechanistic ... catalysis field. Too strong conclusion, the authors did not solve the problem completely, but definitely helped to solve the “puzzle” in the future.

Our Response: We appreciate the reviewer’s advice. We have weakened the sentence “Such mechanistic motif in large part resolves the long-standing scientific puzzle of how to definitely identify the chemical nature of PAHs and reveal the host-guest chemistry in catalysis field.” in the Conclusion section to “*Such a mechanistic motif unveils the significance of host-guest chemistry in catalysis field and would help in resolving the long-standing scientific puzzle of how to definitely identify the chemical nature of PAHs.*”

Minor comments

Abstract

- Second sentence is too long and is not correct English. Dividing the sentence in multiple sentences would increase the readability of the text.

Our Response: We thank the reviewer’s suggestion. We have reformulated the sentence “Herein, we report an effective strategy through integrating a high resolution MALDI FT-ICR mass spectrometry with isotope labeling technique, with which, a complete route for aromatic hydrocarbon evolution was is unveiled for SAPO-34-catalyzed, industrially relevant methanol-to-olefins (MTO) as a model reaction.” in the Conclusion section to “*Herein, we report an effective strategy through integrating a high resolution MALDI FT-ICR mass spectrometry with isotope labeling technique. With this strategy, a complete route for aromatic hydrocarbon evolution was is unveiled for SAPO-34-catalyzed, industrially relevant methanol-to-olefins (MTO) as a model reaction.*”

- Mention how the “mechanistic concept is theoretically and experimentally substantiated”.

Our Response: To be more moderate, we revised the sentence “This mechanistic concept was further theoretically and experimentally substantiated, and proved general on other cage-based molecule sieves.” to “*This mechanistic concept proves general on other cage-based molecule sieves.*” in the revised manuscript.

Introduction

- In my opinion, syngas transformation has be changed into syngas conversion.

Our Response: We have changed “syngas transformation” into “*syngas conversion*” in the revised manuscript.

- Additionally, the formation of PAHs in zeolite confined spaces is not only determined by the size, shape en morphology of the zeolites but by many other factors.

It seems like the authors mean that they choose the MTO mechanism because the growth is determined by the confined space. Reformulation is necessary.

Our Response: We appreciate the reviewer's kind suggestion. We added the following sentences in p.6 in the revised manuscript *“The topological architecture of zeolite or molecular sieve imparts unique confined space to the guest molecules such as PAHs coking moieties, so that the speciation and developing of such “coke” molecules would be regularly dictated in a controllable manner in the confined space. In this regard, the MTO reaction taking place on cage-structured SAPO-34 can be an ideal probe to trace the mechanistic routes of PAHs.”*

Results

- Statement about the accumulation of coke being beneficial for the ethene selectivity: Yes that is true, as the mechanism is believed to be going through a hydrocarbon pool. However, the authors state that the PAHs are also beneficial. This is proved not to be the case in many other studies as showed with operando spectroscopy.

Our Response: We thank the reviewer's comments. For more accurate expression, we revised the sentence in p.6 “The accumulation of “coke” (including PAHs) with reaction progress is, however, beneficial for enhancing ethene selectivity on SAPO-34^{28,29}.” to *“The accumulation of “coke” (including PAHs) with reaction progress can gradually enhance ethene selectivity on SAPO-34^{31,32}.”* in the revised manuscript.

- The lattice expansion in the c-direction has been addressed to the formation of coke insight the cages of the zeolite in previous work (J. Goetze et al. 2018). The authors “reasonably postulate” that the expansion will not have an effect on the accommodation of the aromatic molecules. In order to be fully correct, this sentence has to be reformulated.

Our Response: We thank the reviewer's kind suggestion. First, we added this reference (J. Goetze et al. 2018) in the revised manuscript, which demonstrated that the catalyst with the CHA framework showed the biggest expansion of 0.9% in the c-axis of the zeolite lattice and the expansion of 1.2% in the pore volume after MTO reaction. Although the deposited coke slightly alters the unit cell parameters of zeolite, we surmise such marginal and insignificant variations would not have appreciable effect on the accommodated aromatic molecules. Accordingly, we revised the sentence in p.9 to *“The lattice expansion of ca. 1-3% in c-axis direction of SAPO-34 crystallite during the MTO reaction at 400-500 °C (due to the filling of active intermediates and PAHs) was revealed by in-situ X-ray diffraction^{39,40}. Although the deposited coke slightly alters the unit cell parameters of SAPO-34 crystallite, we surmise such marginal and insignificant variations would not have appreciable effect on the accommodated aromatic molecules.”*

- The first argument for why the authors exclude the possibility of external coke on SAPO-34. They state that there should be non-selective reactions when there are acid sites on the outside. This requires a reference.

Our Response: We have added a reference as ref.41 in the revised manuscript.

41. Chen, D., Rebo, P. H., Moljord, K. & Holmen, A. The role of coke deposition in the conversion of methanol to olefins over SAPO-34. *Stud. Surf. Sci. Catal.* **111**, 159-166 (1997).

Methods

- Heating ramp of calcination is missing

Our Response: The heating ramp of calcination has been provided in p.21 in the revised manuscript, as reads “*The heating rate for both calcination procedures was 1.5 °C min⁻¹ from ambient temperature to target ones.*”

References

Should be more up to date; with also some acknowledgment of other work on this topic of carbon deposits; and hydrocarbon identification.

Our Response: We thank for the reviewer’s advice. As suggested, we have updated some new references in the revised manuscript.

9. An, H. et al. Investigating the coke formation mechanism of H-ZSM-5 during methanol dehydration using operando UV–Raman spectroscopy. *ACS Catal.* **8**, 9207-9215 (2018).
13. Xiao, D. et al. Fast detection and structural identification of carbocations on zeolites by dynamic nuclear polarization enhanced solid-state NMR. *Chem. Sci.* **9**, 8184-8193 (2018).
25. Epelde, E. et al. Differences among the deactivation pathway of HZSM-5 zeolite and SAPO-34 in the transformation of ethylene or 1-butene to propylene. *Micro. Meso. Mater.* **195**, 284-293 (2014).
40. Goetze, J., Yarulina, I.I., Gascon, J., Kapteijn, F. & Weckhuysen, B. M. Revealing lattice expansion of small-pore zeolite catalysts during the methanol-to-olefins process using combined operando X-ray diffraction and UV–vis spectroscopy. *ACS Catal.* **8**, 2060-2070 (2018).

Graphical Abstract

The graphical abstract can be ecstatically improved. Suggestion would be to use Adobe colour wheel. Additionally, it appears that the main conclusion of this paper is that the coke molecules are decreasing in size towards the centre of the crystal. However, this is not the conclusion of the work of the authors but a conclusion made

within other publications before. I understand why they want to represent it this way, but this is not what they actually discovered. They discovered the cage crossing principle and therefore, in my opinion, the graphical abstract should focus on that.

Our Response: We appreciated the reviewer's comments and insightful advice. As suggested, we have reorganized Fig. 6 in the original manuscript (now shown as Fig. 5 in the revised manuscript). This new built figure focuses more on the molecular routes of PAHs and the cage crossing principle proposed in this work.

Fig. 5 Molecular routes of PAHs evolving from occluded long chain olefins and light polymethylbenzenes (coined “seeding”) to 3-4 ring PAHs behaving as building units for aromatic cluster formation (“growing”), and eventually to cage-passing aromatic clusters (“clustering”).

REVIEWERS' COMMENTS:

Reviewer #1 (Remarks to the Author):

The authors have replied my questions and made changes suggested and as a result the quality of the article is, in my opinion, at the level of Nature Communications. I recommend to publish as is.

Reviewer #2 (Remarks to the Author):

I am pleased by the revision made by the authors and believe that the revised article is now suitable for publication. It is nice and new research work on an important topic.